# ROX Index Variation as a Predictor of Outcomes in COVID-19 Patients

**DOI:** 10.3390/jcm13113025

**Published:** 2024-05-21

**Authors:** Augusto Maldonado, Pablo Endara, Patricio Abril, Henry Carrión, Carolina Largo, Patricia Benavides

**Affiliations:** 1School of Medicine, Universidad San Francisco de Quito, Quito 170901, Ecuador; pendara@usfq.edu.ec (P.E.); henryacarrion@hotmail.com (H.C.); 2Hospital General Docente de Calderón, Quito 170201, Ecuador; patoabrillopez@hotmail.com (P.A.); ldcarol.94@gmail.com (C.L.); patricia.benavides@hgdc.gob.ec (P.B.)

**Keywords:** ROX Index, COVID-19, hypoxemic respiratory failure, high-flow nasal cannula, intubation, low resource settings, prediction tools

## Abstract

**Background:** During the COVID-19 pandemic, emergency departments were overcrowded with critically ill patients, and many providers were confronted with ethical dilemmas in assigning respiratory support to them due to scarce resources. Quick tools for evaluating patients upon admission were necessary, as many existing scores proved inaccurate in predicting outcomes. The ROX Index (RI), a rapid and straightforward scoring system reflecting respiratory status in acute respiratory failure patients, has shown promise in predicting outcomes for COVID-19 patients. The 24 h difference in the RI accurately gauges mortality and the need for invasive mechanical ventilation (IMV) among patients with COVID-19. **Methods:** Study design: Prospective cohort study. A total of 204 patients were admitted to the emergency department from May to August 2020. Data were collected from the clinical records. The RI was calculated at admission and 24 h later, and the difference was used to predict the association with mortality and the need for IMV, a logistic regression model was used to adjust for age, sex, presence of comorbidities, and disease severity. Finally, the data were analyzed using ROC. **Results:** The difference in respiratory RI between admission and 24 h is a good predictor for death (AUC 0.92) and for mechanic ventilation (AUC: 0.75). Each one-unit decrease in the RI difference at 24 h was associated with an odds ratio of 1.48 for the risk of death (95%CI: 1.31–1.67) and an odds ratio of 1.16 for IMV (95% IC: 1.1–1.23). **Conclusions:** The 24 h variation of RI is a good prediction tool to allow healthcare professionals to identify the patients who will benefit from invasive treatment, especially in low-resource settings.

## 1. Introduction

The coronavirus disease 2019 (COVID-19) pandemic represented a major global health threat and most health systems were easily overwhelmed [1]. The global mortality rate of COVID-19 deaths is far more than the published data [2]. In its latest report, the WHO stated a mortality rate of greater than 66% [3]. During the first and most devastating wave of COVID-19, Ecuador had one of the highest mortality rates in South America [4].

In low-resource settings, the number and severity of cases resulted in high mortality due to a lack of mechanical ventilators and as well as other medical supplies. The prioritization of patients through triage tools was essential to provide opportunities to those with the best chance of survival and to facilitate decisions regarding their appropriate referral level. During triage, using simple and cost-effective methods is warranted in any situation, especially in low-resource settings [5].

Acute respiratory distress syndrome (ARDS) was present in up to 20% of patients with COVID-19, most of them with an urgent need for mechanical ventilation [6,7] so the development of a rapid prediction tool for COVID-19 severity needed to be established as soon as possible.

In COVID-19, the main concern was the rapid respiratory decompensation that led to establishing early intubation and IVM protocols based on the increased O2 needs [8,9,10]. Identifying the correct moment to intubate is important, as the risk of delaying this procedure leads to severe complications in a short time [11,12,13,14].

One of the challenges emergency physicians confronted during the emergency room visit was to estimate the severity of illness and prognosis [15], ideally completed with simple, precise, and practical scales applied in conditions like this pandemic [16].

The need for early predictors to identify high-risk patients and those with the best chance of survival was a high priority. Many scales have been used but with limitations in their predictive power. Previous tools were often time-consuming and complex to calculate. One possible solution, the RI, described by Roca [17], expresses the relation between pulse oximetry/inspired oxygen fraction (FiO2)/respiratory rate. It is a useful tool in decision-making and in the immediate therapeutic management of patients [17,18]. Originally used for in-patients with pneumonia with acute respiratory failure treated with high flow nasal cannula (HFNC), the RI can help identify those patients with low and those with high risk for intubation [18].

Although the predictive capacity of RI is adequate, it is not high enough to be used as the sole criterion to predict failure of the HFNC, and those who require IMV. Therefore, it is important to add to this tool clinical variables or serial measurements to improve the predictive value.

The objective of the present study was to evaluate the 24 h RI variation as a noninvasive tool in the emergency department to define the possible outcome of COVID-19 patients [12,13] in the pandemic scenario.

## 2. Materials and Methods

### 2.1. Design and Study Population

This is a prospective cohort study. Between 1 May to 31 August 2020, a total of 204 consecutive patients were admitted to the COVID-19 treatment area in a reference second-level hospital in Quito, Ecuador (2850 m altitude). The following inclusion criteria were utilized: age 18 or older, met one or more of the WHO criteria for confirmed or strongly suspicious cases of COVID-19, which are a positive PCR-rt test in the first case, or chest CT (computed tomography) scan result highly suspicious (CORADS) of COVID-19, and clinical–epidemiological criteria for the second one. In the health public system at the time of the study, the availability of COVID-19 molecular diagnostic tests was scarce and limited, and it was therefore impossible to test everyone. Only 25 patients out of 204 patients had the test performed. The CT, on the other hand, was available to every patient with a clinical suspicion of COVID-19.

The protocol was approved by the ethics committee of the hospital MSP-CZ9HGDC-2021-0700-O.

No intervention was made in this research, there is no individual identification data shown, and data were obtained from the clinical records. There was no need for informed consent.

### 2.2. Information Collected

In 204 patients with a high likelihood of COVID-19 infection, respiratory parameters like FiO2, respiratory rate, and O_2_ saturation were recorded, and the RI was calculated at time of presentation to the emergency department and 24 h later. Other variables considered were the presence of comorbidities defined by presence/absence and tomographic compromise defined by the CORADS classification. The main outcomes were death and the need for IMV. Regarding the FiO2 during the pandemic, normal nasal cannulas and oxygen masks with and without reservoirs were available. Estimation of FiO2 was through the flow used for those devices. Regarding nasal cannula, for each liter of oxygen, there was a 4% increment in FiO2 until 40% or a maximum of 5 L of flow. Oxygen mask can provide 28% to 50% and non-rebreathing oxygen mask with reservoir can provide 60–90% depending on flow in each case. Respiratory rate was taken in one minute with the patient lying down. Oxygen saturation level was measured through the monitors available in our hospital.

### 2.3. Analysis Plan

Description of respiratory parameters was made presenting means and standard deviations of each parameter at time of presentation and 24 h later. Differences in respiratory parameters between those who survived and those who deceased or those who needed and did not need IMV were tested using *t*-test with correction for unequal standard deviation when necessary. ROC analysis was performed for each respiratory parameter and the RI difference was used to calculate the AUC, for both defined outcomes of death and the need for IMV. The difference between time of presentation (0 h) and 24 h respiratory RI was explored by subtracting 0 h minus 24 h values. Cut-offs for parameters were obtained by calculating the Youden index for each analyzed parameter. An ROC analysis was performed to generate a logistic regression model adjusting for age treated as continuous variable, sex, presence of comorbidities treated as binary variables (presence/absence), and radiographical disease severity as defined by CORADS score using scores 1 and 2 as lower severity, scores 3 and 4 middle severity, and 5 and 6 higher severities [19,20].

To analyze changes in 24 h RI difference, the RI scores were categorized as: >20 = normal respiratory function; 15–19.9 = mild respiratory deterioration; 10–14.9 = moderate respiratory deterioration; and <10 severe respiratory function deterioration. Logistic regression analysis was used to explore the association between the change in RI using the improving status (change from a worse to better category) as baseline category and compared with those who did not change category, or with those who deteriorated from one category to the next worse level, or with those who deteriorate more than 1 category. The logistic regression model was adjusted by age, sex, comorbidities, and severity as previously indicated. The results were also explored to investigate if the change in RI category was modified by sex, age, presence of comorbidity, and severity. The analysis was performed in STATA software version 16.

## 3. Results

### 3.1. Characteristics of the Population

A total of 204 patients were admitted to the hospital with a COVID-19 diagnosis. The mean age was 57 years and 60% of the patients were male. Eighty-eight patients (43%) had comorbidities, the most frequent being hypertension and diabetes mellitus. Fever, cough, dyspnea, and malaise–myalgia were the most common presenting symptoms. Most patients (93%) had a CT scan highly suspicious of pulmonary infection according to the imagen CORADS classification. Approximately one-quarter of patients required IMV, and 56 patients (27%) died. There was no statistical difference in age or comorbidity type between groups compared to death and IMV requirements. Male patients were more frequent among those who died and in IMV. Symptoms were similar between compared groups, except for dyspnea, which was more frequent in those requiring ventilatory support.

Of the 204 patients, 89.8% arrived at the hospital without supplemental O2, (FiO2 of 0.21) only 11.2% were on oxygen at different concentrations and came by ambulance. After 24 h from the time of presentation, oxygen demands were higher (FiO2 > 0.4) in those who died or needed IMV (Table 1).

### 3.2. Respiratory Associated Factors for Death and Mechanical Ventilation

The oxygen saturation levels were statistically higher at the time of presentation and 24 h later in those patients who survived than those who did not, and in those who required IMV when compared with those who did not. FiO2 administered levels and respiratory rate were statistically higher in deceased patients and in those requiring IMV, both at the time of presentation and after 24 h. RI values were significantly higher at presentation and 24 later in patients who survived and those who did not need IMV. Differences in RI values were negative in survivors and in those who did not require IMV. These differences were statistically higher, showing deterioration of RI, in those who did not survive and those who had IMV (Table 2).

### 3.3. Prediction Analysis of Respiratory Parameters and Difference between RI at Entry versus 24 h Later for Risk of Death and Mechanical Ventilation

The difference in RI at entry versus 24 h is a better predictor for death (AUC 0.92) than for a mechanical ventilation predictor (AUC: 0.75) (Table 3) (Figure 1 and Figure 2). Each decrease in one unit of difference of RI correlated to an increased risk of death and IMV by 48% and 16%, respectively. This yielded an odds ratio for death risk of 1.48 (95%CI: 1.31–1.67) and for IMV, 1.16 (95%IC: 1.1–1.23), adjusted for age, sex, comorbidities, and severity of the disease.

### 3.4. Association between 24 h Changes in Respiratory ROX Index and Death

A greater proportion of patients were admitted with moderate to severe respiratory deterioration (62%), and 24 h later around 39% of patients remained in those categories. Similarly, a minority of patients were admitted with normal respiratory function, but this value increased 24 h later (Table 4).

The higher the deterioration in 24 h in RI score, the higher the likelihood of death or need for IMV. Even in the absence of change in the first 24 h, there is a statistical increase in the likelihood of death and IMV, compared with those who improve RI scores in the first 24 h. The probability of death and IMV increase when the deterioration of RI is even higher. This association was independent of age, sex, disease severity, and presence of comorbidities and there was no modification effect in this association due to these confounders (Table 5).

The baseline category (Table 5) improvement means a change from a worse to a better functional respiratory category. No change refers to no change in the 24 h in the functional respiratory category compared to the initial category. Deteriorating 1 category means a change from normal to mild, mild to moderate, or moderate to severe respiratory deterioration in 24 h. Deteriorating 2 category means a change from mild to severe, or normal to moderate respiratory function in 24 h.

## 4. Discussion

The RI is a valuable tool for quick evaluation of the condition of COVID-19 patients at the time of presentation and it has a good prognostic value (AUC 78.6) to predict mortality and the need for IMV (AUC 76.1). Following stabilization and initial treatment, a 24 h RI variation became an even better prognostic tool with an (AUC 92) and (AUC 75) for mortality and IMV, respectively; this also shows that a 24 h improvement in the functional respiratory category is associated with survival and no need for IMV. Conversely, a 24 h deterioration in RI, or no change in RI is associated with an increasing risk of death and eventual need for IMV support.

In a health disaster like the COVID-19 pandemic [21], conceptually, the healthcare provider’s efforts should be directed to those with the better chance of survival [22], especially in low-resource settings or countries where infrastructure is poor and access to respiratory equipment and supplies like HFNC and non-invasive ventilation was limited. As far as we have searched the literature, there are some publications about the value of the RI in predicting mortality or mechanical ventilation in COVID-19 patients [23,24]. Vega et al. establish that RI is a useful tool to guide intubation, especially in moderate respiratory categories, and Basoulis et al. also show that a 12 h RI is useful in predicting mortality. The 24 h RI difference seems to be a good tool to differentiate between those who will survive and allows for directing resources and therapy in the right way, but maybe this is a long interval to improve or change therapeutic measures. It would be beneficial to calculate the RI difference in shorter intervals to seek a prognostic benefit for individuals. The results of other authors show different values of the RI as predictors of the same outcomes considered in this paper, for example, Vanni et al. found that RI < 5.83 on ED admission was 79.6% sensitive and 63.5% specific for predicting mortality/mechanical ventilation need; and Bartoletti et al. show that an RI of <3.85 predicted mortalities with a sensitivity of 76.9% and a specificity 65.8%, [25,26] a much lower predictive power than the difference in RI at 24 h.

This research was conducted in Quito, Ecuador and 204 patients were enrolled during the critical period of the pandemic. We have identified some limitations with our study. First, this was a monocentric study, second, it was conducted in a public hospital, and therefore patient characteristics might not be similar to other healthcare facilities, so our results cannot be generalizable to all COVID-19 patients. Third, Quito is a high-altitude city (2850 m) and it represents a special environmental situation. The inhabitants of high altitudes usually develop hypocapnic hypoxia and have respiratory rates higher than the sea level, as West B. et al. mention [27], and this fact may have an impact on the RI score and its utilization in other regions. The equation to calculate the RI has three components: oxygen saturation, FiO2, and respiratory rate, the component that remains the same between high altitude and sea level is inspired oxygen fraction (21%) [28,29]. Due to this, we introduced four groups of the RI based on the normal parameters for Quito (Table 4). This may have become a source of bias during the analysis. Gianstefany et al. described sea level values for an RI higher than those in Quito [16], for instance, the RI cut-off for ambulatory care in the Gianstefany research was 26, which is different from 20 in our study. Lastly, the FiO2 was estimated based on clinical practice guidelines [30,31] rather than direct measurement studies. In the pandemic, the exact measurement of the administered FiO2 was not feasible in our setting as we lacked the necessary devices to do it; this could also become a limitation in our research.

We describe four categories to show different stages of respiratory compromise in patients with COVID-19 from normal to severely compromised respiratory function. If patients do not improve the RI score within 24 h after implementing all the initial treatments or if the score deteriorates in one or two categories, the risk of death and need for mechanical ventilation increases. The greater odds ratio values were explained by the small study population in each category. However, even in the small groups, the association between respiratory deterioration and death or IMV was consistent.

Patients with RI at admission superior to 20 or those who improved their RI in 24 h were assigned to observation or discharged for ambulatory treatment.

The in-hospital mortality was 27% (56/204) with a length of hospital stay of 7.27 days on average. The average RI in this group deteriorated from 9.8 to 4.9, showing the rapid progression and severity of the disease. Twelve patients died at home after being discharged from the hospital. Ten of these patients improved the 24 h RI difference and only two deteriorated. Perhaps this group of patients died because of complications of the disease but this analysis is out of the scope of this report.

## 5. Conclusions

The use of the ROX Index at the time of admission for COVID-19 pneumonia patients is an easy and quick method used to predict the severity and outcome of patients when used in emergency departments. Our study shows that its use yields the same or more accurate prediction of outcomes compared to other similar scales. Interestingly, the 24 h difference in RI has the greatest potential to predict the outcome of patients and help identify those who will benefit from intensive measures.

More research on this topic is needed to understand and implement the use of this tool, especially in low-resource settings where equipment is scarce, and infrastructure is poor.

## Figures and Tables

**Figure 1 jcm-13-03025-f001:**
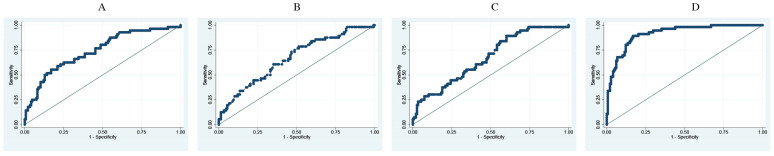
ROC curves for death prediction. Adjusted for age, sex, comorbidities, and disease severity. (**A**). Sat O_2_. (**B**). FiO_2_. (**C**). Respiratory frequency. (**D**). ROX Index difference.

**Figure 2 jcm-13-03025-f002:**
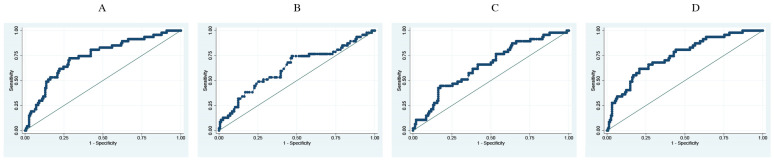
ROC curves for mechanic ventilation prediction. Adjusted for age, sex, comorbidities. (**A**). Sat O2. (**B**). FiO2. (**C**). Respiratory frequency. (**D**). Respiratory ROX Index difference.

**Table 1 jcm-13-03025-t001:** Characteristics of studied population by main outcomes.

		Death		*p*-Value	Mechanic Ventilation		*p*-Value
		*n* = 148	*n* = 56		*n* = 157	*n* = 47	
		No	Yes		No	Yes	
Age	Mean; SD	55.8; 1.4	60.6; 2.1	0.06	57.6; 1.4	55.6; 1.8	0.47
Sex	Male	83 (56.1)	40 (71.4)	0.05	89 (56.7)	34 (72.3)	0.05
Comorbidities	Diabetes mellitus	20 (13.5)	8 (14.3)	0.88	23 (14.7)	5 (10.6)	0.48
	Hypertension	33 (22.3)	11 (19.6)	0.68	34 (21.7)	10 (21.3)	0.95
	Lung disease	6 (4.1)	3 (5.4)	0.68	5 (3.2)	4 (8.5)	0.12
	Hipotiroidism	6 (4.1)	3 (5.4)	0.68	5 (3.2)	4 (8.5)	0.12
	Renal disease	6 (4.1)	3 (5.4)	0.68	5 (3.2)	4 (8.5)	0.12
	Surgical history	5 (3.4)	0	na	5 (3.2)	0	0.21
	Cerebral vascular disease	2 (1.35)	1 (1.8)	0.81	2 (1.3)	1 (2.1)	0.7
	Obesity	3 (2)	2 (3.6)	0.52	3 (1.9)	2 (4.3)	0.36
	Any comorbidity	66 (44.6)	21 (37.5)	0.36	67 (42.7)	20(42.5)	0.98
Symptoms/signs	Fever	74 (50)	32 (57.1)	0.36	75 (47.8)	23 (49)	0.89
	Cough	97 (65.5)	31 (55:4)	0.18	97 (61.8)	31 (65.9)	0.6
	Dyspnea	103 (69.6)	43 (76.8)	0.31	107 (68.1)	39 (83)	0.05
	Myalgias	68 (45.9)	25 (44.6)	0.86	77 (49)	16 (34)	0.07
CORADS	Grade 1–2	2 (1.4)	1 (1.8)		3 (1.9)	0	
	Grade 3–4	9 (6.1)	2 (3.5)		9 (5.7)	2 (4.3)	
	Grade 5–6	137 (92.6)	56 (94.7)	0.76	145 (92.4)	45 (95.7)	0.6
FiO2 at Admission	0.21	135 (91.2)	46 (82.1)		142 (90.5)	39 (83)	
	0.22–0.4	12 (8.1)	4 (7.1)		13 (8.3)	3 (6.4)	
	>0.4	1 (0.7)	6 (10.8)	0.02	2 (1.2)	5 (10.6)	0.008
FiO2 at 24 h	0.21	95 (64.2)	0		89 (56.7)	6 (12.8)	
	0.22–0.4	49 (33.1)	6 (10.7)		51.32.5)	4 (8.5)	
	>0.4	4 (2.7)	50 (89.3)	<0.0001	17 (10.8)	37 (78.7)	<0.0001

**Table 2 jcm-13-03025-t002:** Associated factors for death and mechanical ventilation.

	Death			Mechanic Ventilation		
	Survived	Deceased	*p* Value	Absent	Present	*p* Value
	*n* = 148	*n* = 56		*n* = 157	*n* = 47	
Age: mean; SD	55.8; 16.5	60.5; 15.9	0.06	57.6; 17.3	55.6; 12.8	0.47
Disease time: mean; SD	7.3; 4.7	7.5; 5.3	0.72	7.15;5.1	7.9; 4.26	0.34
Hospitalization time: mean; SD	9.1; 9.8	10.3; 10	0.46	6.9 (5.8)	18.3; 14.7	<0.0001
Sat O2 entry: mean; SD	79.7; 0.95	67.7; 19.8	0.0001	78.8; 13.3	68.4; 18.3	<0.0001
Sat O2 24 h: mean; SD	91.6; 2.2	82.4; 10.4	<0.0001	89.9; 6.7	86.2; 7.6	0.0016
Difference sat O2: mean; SD	−11.9; 11.3	−14.6; 22.1	0.37	−11.1; 13.2	−17.8; 19.2	0.007
FiO2 admission, mean; SD	0.22; 0.05	0.27; 0.17	0.04	0.22; 0.07	0.26; 0.16	0.02
FiO2 24 h, mean; SD	0.24; 0.1	0.70; 0.23	<0.0001	0.29; 0.18	0.62; 0.27	<0.0001
Resp. Freq. admission	27.6; 7.6	31.6; 8.9	0.004	27.9; 7.7	31.3; 9.1	0.01
Resp. Freq. 24 h	22.7; 6.2	30.2; 7.8	<0.0001	23.5; 5.7	28.8; 10.6	<0.0001
ROX Index admission	14.3; 4.5	9.8: 4.8	<0.0001	14; 4.9	10; 4.2	<0.0001
ROX Index 24 h	18.54; 4.2	4.99; 2.99	<0.0001	17.1; 6	7.4; 5.8	<0.0001
Difference ROX Index mean; SD	−4.3; 5.3	4.8: 4.9	<0.0001	−3.06; 6.13	2.6; 6.2	<0.0001

**Table 3 jcm-13-03025-t003:** Predictors of death and mechanical ventilation.

Parameter	Death			Mechanic Ventilation		
	AUC	Sensitivity	Specificity	AUC	Sensitivity	Specificity
O2 sat admission	0.7410	60.71	77.70	0.7352	72.34	69.42
FiO2 admission	0.6761	62.5	64.19	0.6237	76.59	53.5
Resp. Freq admission	0.6792	91.07	39.86	0.6434	76.59	53.5
ROX Index	0.7868	75	72.97	0.7616	74.47	70.06
Difference in ROX Index	0.92	0.89	0.80	0.75	57	83

**Table 4 jcm-13-03025-t004:** Frequency of respiratory severity categories at entry and 24 h later.

Functional Respiratory Category	Admission	24 h
	*n* (%)	*n* (%)
≥20 Normal	17 (8.3)	54 (26.5)
15 to 19.9 Mild	60 (29.4)	71 (34.8)
10 to 14.9 Moderate	64 (31.4)	24 (11.8)
<10 Severe	63 (30.9)	55 (26.9)

**Table 5 jcm-13-03025-t005:** Association between 24 h ROX Index change and death or mechanic ventilation.

	Death				Mechanic Ventilation			
ROX Index 24 h change	Survive	Death	Crude OR (95%CI)	Adj. OR (95%CI)	no MV	MV	Crude OR (95%CI)	Adj. OR (95%CI)
Improvement	87 (97.7)	2 (2.3)	1	1	81 (91)	8 (9)	1	1
No change	46 (58.2)	33 (41.8)	31.2 (7.2–135.9)	46 (9.8–216.2)	55 (69.6)	24 (30.4)	4.4 (1.9–10.5)	4.4 (1.8–10.7)
Deteriorating 1 category	14 (53.8)	12 (46.2)	37.3 (7.5–184.6)	38 (7.1–199.8)	17 (65.4)	9 (34.6)	5.4 (1.8–15.9)	5.2 (1.71–15.6)
Deteriorating 2 categories	1 (10)	9 (90)	391 (32.2–4753.4)	674 (47.5–9564)	4 (40)	6 (60)	15.2 (3.5–65.3)	12.9 (2.9–56.5)

## Data Availability

The data presented in this study are available on request from the corresponding author.

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
