# Peer review of "ROX Index Variation as a Predictor of Outcomes in COVID-19 Patients"

_jcm, 2024, doi:10.3390/jcm13113025_

Round 1
Reviewer 1 Report
Comments and Suggestions for Authors
Dear authors,
Thank you for the opportunity to review your work on the Rox Index Variation as a predictor of outcomes in patients admitted to a hospital in Quito during the Covid-19 pandemic. I think you took on a project which is clinically relevant, and is of particular interest in triage medicine of respiratory conditions. I also think your methodology was good and I believe your conclusions are probably supported by your data, but there are several points of concern which I would like to raise:
Major issues:
1. It is unclear to me how the FiO2 for these patients was calculated. I assume that these patients were on various combinations of nasal cannula, high flow nasal cannula, and BiPAP, but your paper does not discuss how you ascertained their FiO2. This needs to be described very clearly, since it is a key component of your primary outcome. Your Table 1 should also very clearly describe what type of supplemental oxygen they were receiving.
2. Progression of Covid-19 severity is associated with comorbid medical conditions. While you do describe the comorbidities seen in your patient population, you do not provide a breakdown of these comorbidities as related to the groups who died and those who didn't, and those who required mechanical ventilation, and those who didn't. I would recommend that you restructure your Table 1 to have these subheadings. It is important for the readers to know whether there were significant differences in baseline oxygen use and comorbidities. In particular, how many patients had underlying COPD or ILD? How many patients had altitude-associated pulmonary hypertension?
3. Related to #2, you make the statement that you adjusted OR for age, sex, comorbidity, and severity of disease, but you don't describe in your methods how this was done. For example, did you adjust for older age vs younger age, or as a continuous variable? Did you adjust for each comorbidity individually, or based on a scoring system? I might suggest that you use the Elixhauser index, which is a standardized scoring system of comorbidities in acute illness.
Minor issues:
1. The tables should be cleaned up. There are multiple places where the columns and rows do not line up and it is hard to tell what relates to what.
2. The English needs a lot of work. Please see comments below.
3. The Methods section needs work. This seems like a prospective cohort study, but you do not explicitly describe it as such. Some of the inclusion criteria are vague. For example, in lines 106-107 you mention that either rt-PCR or CT chest was used to diagnose Covid-19. Since CT chest for Covid-19 can look a lot like pulmonary edema (or a variety of other pulmonary illnesses), how do you know that these patients truly had Covid-19? Furthermore, in your results you should describe how many patients had a positive rt-PCR and how many were diagnosed on the basis of CT scans.
4. I think the Introduction is a bit long and difficult to follow. You have the right information, but I would edit it to have a more linear structure, and potentially move some of the components into the discussion.
Comments on the Quality of English LanguageI would strongly recommend that this paper be reviewed by someone with native-level command of English and experience in medical publication. There are many phrases which seem translated from Spanish which are awkward in English. For example, line 135: "This arbitrary classification is based on normal values for Quito altitude." What does this mean? Are there validated RI values for Quito? " Another example is in line 187: "The higher the deterioration in 24 hs in RI score, the higher the likelihood of death or need for mechanic support." This is not syntactically correct, and also unclear what is meant by "mechanic support" - are you referring to intubation or NIPPV? Finally, there are multiple spelling errors (for example, "Deterioring" in Table 5) and in line 238 in the words "OR for deaths was 38 y 674," "y" was not translated into "and." There are several such places in the paper.
I strongly encourage and commend researchers from non-English speaking countries, but accuracy of spelling and grammar enhances the legitimacy of your work.
Author Response
We are very grateful and pleased by the reviewer´s comments. We found them very useful Here, the answers and considerations to improve the manuscript.
Major issues:
- It is unclear to me how the FiO2 for these patients was calculated. I assume that these patients were on various combinations of nasal cannula, high flow nasal cannula, and BiPAP, but your paper does not discuss how you ascertained their FiO2. This needs to be described very clearly, since it is a key component of your primary outcome. Your Table 1 should also very clearly describe what type of supplemental oxygen they were receiving.
Answers:
The inspired oxygen fraction (FiO2): During the pandemic, we had normal nasal cannulas, oxygen masks with and without reservoirs (non-rebreathing mask). Calculation of FiO2 was through the flow used for those; for nasal cannula each liter of oxygen flow represented a 4% increment in FiO2 until 40% or a maximum of 5 lt. Regarding to oxygen mask you can provide 28% to 50% and oxygen mask with reservoir 60%-90% depending on flow in each case. Respiratory frequency was taken in one minute with the patient lying down. Oxygen saturation was measured through the monitors available in our hospital.
We have incorporated a new paragraph about the (FiO2) under the subheading information collected in the Material and Methods section.
Table 1 has been restructured to show frequencies of patients with FiO2 of 0.21, or 0.22-0.4 and >0.4, according with death and mechanic ventilation groups as well as for comorbidities.
- Progression of COVID-19 severity is associated with comorbid medical conditions. While you do describe the comorbidities seen in your patient population, you do not provide a breakdown of these comorbidities as related to the groups who died and those who didn't, and those who required mechanical ventilation, and those who didn't. I would recommend that you restructure your Table 1 to have these subheadings. It is important for the readers to know whether there were significant differences in baseline oxygen use and comorbidities. In particular, how many patients had underlying COPD or ILD? How many patients had altitude-associated pulmonary hypertension?
Answers:
Thank You for this comment we are providing the information that neither of the comorbidities had a statistic association with mortality and or mechanical ventilation. Also, table 1 has been restructured to show comorbidity types by death and mechanic ventilation. Unfortunately, we do not have specific information in database about the type of disease and complications for example COPD and High-altitude Pulmonary Hypertension as you ask for. Also, the comorbidities were treated in a binary way (any comorbidity) for easier way to include in logistic regression model.
- Related to #2, you make the statement that you adjusted OR for age, sex, comorbidity, and severity of disease, but you don't describe in your methods how this was done. For example, did you adjust for older age vs younger age, or as a continuous variable? Did you adjust for each comorbidity individually, or based on a scoring system? I might suggest that you use the Elixhauser index, which is a standardized scoring system of comorbidities in acute illness.
Answers:
We have included in the methodology section a phrase to explain how the variables were treated. Age was treated as continuous variable, presence of comorbidities was treated as binary variable based in the presence/absence of any comorbidity, severity was measured based in CORADS classification: 1-2: mild; 3-4: moderate 5-6: severe as it’s shown in the next references:
Bai HX, Hsieh B, Xiong Z, et al. Performance of radiologists in differentiating COVID-19 from viral pneumonia on chest CT. Radiology. 2020;296(2):E46-E54.
Garibaldi BT, Fiksel J, Johnson BA, et al. Severity of COVID-19 respiratory illness and imaging correlation. Nat Commun. 2020;11(1):5333.
Minor issues:
- The tables should be cleaned up. There are multiple places where the columns and rows do not line up and it is hard to tell what relates to what.
Answers:
Thanks for the observation now is done, we have centred and organized the information.
- The English needs a lot of work. Please see comments below.
Answer:
The document has been reviewed and modified by an English native speaker.
- The Methods section needs work. This seems like a prospective cohort study, but you do not explicitly describe it as such. Some of the inclusion criteria are vague. For example, in lines 106-107 you mention that either rt-PCR or CT chest was used to diagnose Covid-19. Since CT chest for Covid-19 can look a lot like pulmonary edema (or a variety of other pulmonary illnesses), how do you know that these patients truly had Covid-19? Furthermore, in your results you should describe how many patients had a positive rt-PCR and how many were diagnosed on the basis of CT scans.
Answers:
Thanks for your commentary, yes this is prospective cohort study, and now this is included in the methods section.
The inclusion criteria: Our team took the WHO criteria for confirmed and suspected case, in the beginning of the pandemic. The PCR-RT tests were very limited in our country and Hospital and imposible to test every suspicious patient, we used a combination of CT findings based in the CO-RADS classification that had excellent sensitivity and specificity to identify patients with COVID 19 and the clinical and epidemiological criteria.
The CORADS also offered the possibility to rule out other diagnosis like cardiogenic acute pulmonary edema with a reasonable safety even more during the pandemia.
- I think the Introduction is a bit long and difficult to follow. You have the right information, but I would edit it to have a more linear structure, and potentially move some of the components into the discussion.
Answers:
Thanks again for your observation, we have made some improvement and cut some paragraphs align better the ideas and improve grammar and syntax to make it more understandable.
Reviewer 2 Report
Comments and Suggestions for Authors
It is an honor to review meaningful research to find out the effectiveness of evaluating and managing emergency patients using Rox Index.
The introduction is properly structured. However, no prior research analysis has been conducted on predictive tools such as rox index. Please add prior research analysis on RI.
Research methods have also been properly structured.
Research results have shown that rox index helps to check the condition of the disease and treat it. Research results have been properly structured.
In the discussion, there was a lack of value and prior research on the rox index, so please conduct the discussion with previous studies on various rox indices through more search. The current discussion is reaching the level of simple interpretation of the results.
It seems to be a meaningful study to explore the value of rox index and confirm its potential. However, the paper format is not right overall, and there seem to be many errors in the citation of references. There are no major errors in the research topic and results, but please make the editing as an article a little more delicate.
Author Response
We are very grateful and pleased by the reviewer´s comments. We found them very useful Here, the answers and considerations to improve the manuscript.
The introduction is properly structured. However, no prior research analysis has been conducted on predictive tools such as rox index. Please add prior research analysis on RI.
Answers:
Thanks again for your observation, we have made some improvement and cut some paragraphs align better the ideas and improve grammar and syntax to make it more understandable.
We have added a new paragraph in the discussion section regarding previous research about the predictive value of ROX Index.
Research methods have also been properly structured.
Research results have shown that rox index helps to check the condition of the disease and treat it. Research results have been properly structured.
Thanks for your comments.
In the discussion, there was a lack of value and prior research on the rox index, so please conduct the discussion with previous studies on various rox indices through more search. The current discussion is reaching the level of simple interpretation of the results.
Answers:
Thanks for the advice to include research on Rox Index for COVID-19 patients, some studies have been included with their respective prediction values of Rox Index in the discussion section.
It seems to be a meaningful study to explore the value of rox index and confirm its potential. However, the paper format is not right overall, and there seem to be many errors in the citation of references. There are no major errors in the research topic and results, but please make the editing as an article a little more delicate.
Answer:
Thank You for your kind comment, we have sent the manuscript for a complete review by a native English speaker with experience in medical research.
Reviewer 3 Report
Comments and Suggestions for Authors
- remove the name of the hospital from the title.
- replace "covid-19" with "COVID-19" throughout the manuscript.
- Some abbreviations are not defined in their first use (abstract and main text). Define abbreviations in their first use and make sure that abbreviated forms are being used after the definition.
- The introduction is too long and hard to follow. Only keep essential sentences for background, knowledge gap, and aim. I suggest 2-3 paragraphs for this section. Use previous systematic reviews and mention other biomarkers (endocan, galectins, syndecan) of outcomes in COVID-19. Compare your results with these biomarkers in the discussion section.
- Add software you used for data analysis and visualization.
- Add the strengths and limitations of your study before the conclusions section.
Comments on the Quality of English Language- native review is strongly recommended for typos and grammatical errors.
Author Response
We are very grateful and pleased by the reviewer´s comments. We found them very useful Here, the answers and considerations to improve the manuscript.
Remove the name of the hospital from the title.
Answer:
Thank You. The name of the Hospital has been removed from the Title.
Replace "covid-19" with "COVID-19" throughout the manuscript.
Answer:
We have replaced as suggest.
Some abbreviations are not defined in their first use (abstract and main text). Define abbreviations in their first use and make sure that abbreviated forms are being used after the definition.
Answer:
Thanks for this observation, we have checked all the text that need abbreviations in the first appearance and are done.
The introduction is too long and hard to follow. Only keep essential sentences for background, knowledge gap, and aim. I suggest 2-3 paragraphs for this section. Use previous systematic reviews and mention other biomarkers (endocan, galectins, syndecan) of outcomes in COVID-19. Compare your results with these biomarkers in the discussion section.
Answer:
Thanks for this advice, we have restructured the introduction section and is easy to read and follow, we didn’t use other biomarkers because our aim was to establish an easy way to predict outcomes in low resource settings. The availability of those biomarkers is hard to obtain.
Add software you used for data analysis and visualization.
Answer
Done, we have included the software used in the analysis plan.
Add the strengths and limitations of your study before the conclusions section.
Answer
We have included the limitations of our research as part of the discussion section.
Round 2
Reviewer 1 Report
Comments and Suggestions for Authors
Dear authors,
Thank you for the opportunity to review your paper again. I appreciate the time and effort spent to make the manuscript better. I do think the revised manuscript is a substantial improvement over the original version.
I appreciate the more detailed description of how you arrived at FiO2 among your study population; however, I continue to be unclear at how you calculated your values. For example, how did you conclude that for each liter of oxygen delivered by nasal cannula, the FiO2 increases by 4%? I'm unaware of any validated data which supports that estimate - if there is, please refer to it in your manuscript.
Second, while I generally agree that you can provide around 28-50% with a facemask and around 60-90% with a reservoir, I still don't fully understand how you could arrive at FiO2 - for example, exactly how much FiO2 would someone be receiving at 8 L/min? Or at 10 L/min?
To me, this is a major issue because it represents the cornerstone of your study. I would recommend that you include a table where the reader can review a given patient's FiO2 at whatever amount of oxygen they were receiving. I suggest you review previously published work on this topic, including (for example), Coudroy et al, 2020 (https://doi.org/10.1136/thoraxjnl-2020-214863).
I appreciate your attention to the remainder of my points, which I think have been satisfactorily addressed. However, I think the readers of your manuscript need to be explicitly clear about how you determined FiO2 since it is a key component of the Rox index.
Furthermore, I recommend you address the issue of entrainment of room air - that is, if a patient was very tachypneic and breathing primarily with their mouth while wearing a nasal cannula, one might suspect that a substantial amount of room air (FiO2 21%) might be mixed with the air coming from a nasal cannula. This might be an unavoidable limitation of your study, but it will be a natural criticism and I recommend you mention it in your manuscript.
I hope that my comments were helpful and taken in the constructive light with which they are intended.
Comments on the Quality of English LanguageApart from occasional minor issues with sentence-verb agreement and unusual comma usage, the quality of English has been much improved and minor issues can be resolved by the copyediting team of the journal.
Author Response
Your Commentary:
I appreciate the more detailed description of how you arrived at FiO2 among your study population; however, I continue to be unclear at how you calculated your values. For example, how did you conclude that for each liter of oxygen delivered by nasal cannula, the FiO2 increases by 4%? I'm unaware of any validated data which supports that estimate - if there is, please refer to it in your manuscript.
Second, while I generally agree that you can provide around 28-50% with a facemask and around 60-90% with a reservoir, I still don't fully understand how you could arrive at FiO2 - for example, exactly how much FiO2 would someone be receiving at 8 L/min? Or at 10 L/min?
Dear Reviewer:
Thank you for your insightful review, we are learning much more about this topic over your comments:
We agree that there is no validated data about the 4% increase in FiO2 for each liter of O2 administered. This is based on predictive algorithms of the equipment and instruments used for this purpose. It is also clear that multiple studies have tried to arrive at a formula that can accurately determine the true FiO2 we are administering to our patients, and there are several formulas with varying accuracy, but a definitive consensus to predict the true FiO2 being delivered has not yet been achieved.
We used the estimated FiO2 that can be delivered by different device types according to the flow rate. This is based on some publications that have been established through clinical practice guidelines rather than direct measurement studies. Exact measurement of the administered FiO2 is not possible in our setting as we lack the necessary elements, even worse under the pandemic, the study's purpose was to find a sufficiently sensitive, rapid, and non-invasive indicator to predict prognosis.
We rely on some medical practice guidelines and texts, such as those cited below, which approximately indicate the FiO2 according to the device and flow rate used:
Fuentes S, Chowdhury YS. Fraction of Inspired Oxygen. [Updated 2022 Nov 29]. In: Stat Pearls [Internet]. Treasure Island (FL): Stat Pearls Publishing; 2024 Jan-. Available from: https://www.ncbi.nlm.nih.gov/books/NBK560867/
American Heart Association. 2013. ACLS for experienced providers. Dallas, TX: American Heart Association. Chapter 6 p44.
O’Driscoll BR, Howard LS, Earis J, et al. BMJ Open Resp Res 2017;4:e000170. doi:10.1136/bmjresp-2016-000170
Hardavella G, Karampinis I, Frille A, Sreter K, Rousalova I. Oxygen devices and delivery systems. Breathe (Sheff). 2019 Sep;15(3):e108-e116. Doi: 10.1183/20734735.0204-2019. PMID: 31777573; PMCID: PMC6876135.
We have carefully read the recommended article. It provides an interesting perspective, revealing that a more precise calculation of FiO2 with approximation formulas is likely required. It implicitly proposes a future research field and shows that the effective FiO2 variability is influenced by other factors such as the person's height, respiratory rate, PaCO2, and whether they breath with an open or closed mouth.
It’s also interesting that in the document of the Australian Thoracic Society, they mention that deterioration of patients could be predicted by an increase in respiratory rate and growing oxygen flow or FiO2. But there is also a recognized the variability of oxygen supply through the low flow systems.
Barnett A, Beasley R, Buchan C, Chien J, Farah CS, King G, et al. Thoracic Society of Australia and New Zealand Position Statement on Acute Oxygen Use in Adults: ‘Swimming between the flags. Respirology. 2022;27: 262–76. https://doi.org/10.1111/resp.14218.
To estimate the FiO2 the patients were receiving we use a linear proportion in each of the flow devices like in the table, but we don’t think this table is suitable for the manuscript.
Todur PM, Chaudhuri S, Eeshwar MV, Teckchandani D, Venkateswaran R. Oxygen sources and delivery devices: Essentials during COVID‐19. Indian J Respir Care 2021;10:171‐81.
We think that the primary objective of the study is to compare the variability of the ROX Index at admission and 24 hours, based on the principle that greater severity and lung injury requires higher oxygen demand to maintain oxygen saturation, and increased respiratory rate is also considered a response to severity. When analyzing the variables separately, they did not have the same predictive power as the ROX Index.
We consider it very important to clarify the following:
- We will change the word "CALCULATED FiO2" to "ESTIMATED FiO2".
- In the DISCUSSION section, we will introduce a more detailed explanation of the limitations of the estimated FiO2 and its multiple causes of variation.
Thanks again for your huge feedback, we sincerely appreciate.
Reviewer 3 Report
Comments and Suggestions for Authors
Thanks.
Author Response
Dear Reviewer, thanks. We did not find any comment so we assume that you agree with the last version already sent.